# Artificial Intelligence Assisted Mid-Infrared Laser Spectroscopy In Situ Detection of Petroleum in Soils

**Nataly J. Galán-Freyle** [1], **María L. Ospina-Castro** [2], **Alberto R. Medina-González** [3],
**Reynaldo Villarreal-González** [4], **Samuel P. Hernández-Rivera** [5] **and**
**Leonardo C. Pacheco-Londoño** [1,*]

1. School of Basic and Biomedical Science, Universidad Simón Bolívar, Barranquilla 080002, Colombia; nataly.galan@unisimonbolivar.edu.co
2. Grupo de Investigación Química Supramolecular Aplicada, Programa de Química, Universidad del Atlántico, Barranquilla 080001, Colombia; mariaospina@mail.uniatlantico.edu.co
3. Morrissey College of Arts and Sciences, Boston College, Chestnut Hill, MA 02467, USA; albertomedgonza@gmail.com
4. Grupo de Investigación en Gestión de la Innovación y la Tecnología, Universidad Simón Bolívar, Barranquilla 080002, Colombia; rvillarreal2@unisimonbolivar.edu.co
5. ALERT DHS Center of Excellence for Explosives Research, Department of Chemistry, University of Puerto Rico, Mayagüez, PR 00681, USA; samuel.hernandez3@upr.edu
* Correspondence: leonardo.pacheco@unisimonbolivar.edu.co; Tel.: +57-304-648-9549

**Abstract:** A simple, remote-sensed method of detection of traces of petroleum in soil combining artificial intelligence (AI) with mid-infrared (MIR) laser spectroscopy is presented. A portable MIR quantum cascade laser (QCL) was used as an excitation source, making the technique amenable to field applications. The MIR spectral region is more informative and useful than the near IR region for the detection of pollutants in soil. Remote sensing, coupled with a support vector machine (SVM) algorithm, was used to accurately identify the presence/absence of traces of petroleum in soil mixtures. Chemometrics tools such as principal component analysis (PCA), partial least square-discriminant analysis (PLS-DA), and SVM demonstrated the effectiveness of rapidly differentiating between different soil types and detecting the presence of petroleum traces in different soil matrices such as sea sand, red soil, and brown soil. Comparisons between results of PLS-DA and SVM were based on sensitivity, selectivity, and areas under receiver-operator curves (ROC). An innovative statistical analysis method of calculating limits of detection (LOD) and limits of decision (LD) from fits of the probability of detection was developed. Results for QCL/PLS-DA models achieved LOD and LD of 0.2% and 0.01% for petroleum/soil, respectively. The superior performance of QCL/SVM models improved these values to 0.04% and 0.003%, respectively, providing better identification probability of soils contaminated with petroleum.

**Keywords:** mid-infrared (MIR) laser spectroscopy; quantum cascade lasers (QCLs); artificial intelligence (AI); chemometrics; multivariate analysis; petroleum; soil

## 1. Introduction

One of the most significant environmental problems related to petroleum and its derivatives that modern society currently faces is oil spills. Petroleum escapes occur as frequently on land as they do on oceans. These spills can be caused by natural disasters, personal accidents, and deliberate acts by terrorists [1–6]. Possible causes for inland oil spills include pipeline leaks, road-tanker accidents, insufficient bonding, aircraft accidents, wars, and conflicts. It is therefore essential to develop

methodologies to detect petroleum in soil [7–14] and water bodies [15–18] to assess damage and develop solutions for places contaminated by petroleum spills [19–21].

The particular soil type such as sand, silt, and clay and the amount of organic matter present determine the fate of petroleum hydrocarbons and the extent of damage to vegetation [22]. Serrano-Guzmán et al. [23] reported that "pollution by petroleum hydrocarbons exerts adverse effects on plants indirectly, generating toxic minerals in the soil available for being absorbed, moreover, leads to a deterioration of the soil structure; loss of organic matter content; and loss of mineral nutrients from the soil, such as potassium, sodium, sulfate, phosphate, and nitrate". Moreover, soils are exposed to leaching and erosion. The presence of these pollutants has resulted in the loss of soil fertility, low crop yields, and possible harmful consequences to humans and the entire ecosystem [24].

The current methods used for the detection of petroleum residues in these areas include spectroscopic and chromatographic techniques such as gas chromatography, mass spectrometry, and hyphenated separation/detection techniques. However, these techniques do not allow real-time measurements [25] and require long detection times, which is a critical factor for addressing the problems associated with pollution. In a search for alternatives, more efficient, rapid analysis techniques that can overcome this challenge, such as mid-infrared (MIR) laser spectroscopy using quantum cascade lasers (QCLs) have been found as a possible solution [26]. These devices are characterized by room-temperature operation, continuous-wave operation, high output power, high wall-plug efficiency [27], small footprints, long lifetimes, low energy consumption, and high long-term power stability [28]. Another characteristic of QCLs is the possibility of widely tuning their output frequencies [29].

The main objective of this study was to employ MIR laser spectroscopy using a QCL source to detect insoluble pollutants such as petroleum and its derivatives in soils by simulating contaminated areas. Matrices such as powdered mixtures and soils, in particular, are not easy to analyze by the use of back-reflection infrared spectroscopy in active mode [30–33]. Diverse soil types with various particle sizes were tested in this study. These solid materials show high coefficients of absorption in the MIR region, thus preventing the ready detection of pollutants. This makes contactless in situ acquisition difficult using conventional thermal sources because the sample absorbs almost completely the MIR light, and the amount of back-reflected light is minimal. Therefore, measurements of diffuse reflectance using MIR generate weak signal intensities and, consequently, low signal to noise ratios (S/N). This is affected if the analysis is carried out in a remote sensing modality [34,35]. A quadratic decrease in the signal with distance and angular dependence is experienced [33,36,37]. A QCL emits a markedly more significant number of photons per unit wavelength in comparison to a globar source. This increases the reflected light and the S/N.

A prospective application of this technique is for the detection of crude oil in soils, which represents a severe biodiversity risk. A fundamental advantage of the MIR laser spectroscopic technique is its portability and capability for rapid analysis because the equipment can be transported to the contaminated site. Besides, it is an efficient technique for analyzing complex mixtures such as soils since it overcomes the challenges faced by other traditional methods.

On the other end, several applications of artificial intelligence (AI) for real-time control of pollution minimization and mitigation processes have been recently reported. They demonstrate an emerging area for more extensive studies [38,39]. AI is a rapidly evolving area that offers sophisticated and advanced approaches capable of addressing complicated and challenging problems. Transferring human knowledge into analytical models and learning from data is a task that can be accomplished by soft-computing methodologies [39]. Recently, some applications of AI for real-time control of pollution minimization and mitigation processes have been reported. They demonstrate an emerging area for more extensive studies [38]. AI is a rapidly evolving area that offers sophisticated and advanced approaches capable of addressing complicated and challenging problems. Transferring human knowledge into analytical models and learning from data is a task that can be accomplished by soft-computing methodologies [39].

Support vector machine (SVM) is a technique for modeling classes that includes the concept of learning. SVMs are supervised classification methods based on statistical learning theory. SVMs effectively solve problems of small sample sizes, high dimensionality, and nonlinearity in the learning process. SVMs aim to find an optimal hyperplane that maximizes the margins between those classes which have distinct features, in some instances. Currently, SVMs are widely used in recognition, facial recognition, fault classification [40–42], time series prediction [43], nonlinear system modeling and identification [44,45], water quality event detection [46], and other fields. The use of SVMs has achieved good results [47]. SVMs have the advantage of avoiding data overfitting, which is a disadvantage of other multivariate pattern recognition methods [48].

On the other hand, traditional multivariate analysis (MVA) pattern recognition methods such as principal component analysis (PCA) and partial least squares coupled to discriminant analysis (PLS-DA), are used in conjunction with spectroscopic methods and are widely reported in the literature [33,37,49,50]. PCA is a non-supervised MVA statistical technique. In the case of unknown identities and quantities, PCA is employed to sort these unknowns and to determine underlying information from multivariate raw data. PCA is often utilized to reduce the dimensionality of spectroscopic data with a linear transformation of some principal components (PCs).

PLS-DA is a supervised MVA statistical technique used to group objects into classes. PLS-DA can find latent variables (LVs) with high covariance values in the dependent (y) variable. This MVA pattern recognition method attempts to identify directions and variables in multivariate space, i.e., the LVs. In the case of two classes, 0 is used to denote Class A, and 1 indicates Class B. Using a threshold defined by the operator, the separation between the classes should be clear and linear. This implies that in the optimum separation into classes, Class A should be on one side of the threshold and Class B on the other side of it. In the case where the predictions provide no clear distinction between the classes, the number of false positives increases and the model is not suitable for discrimination [33,37,49,50].

In this study, QCL assisted with the recent advancements of AI-based technologies that were evaluated for the detection of petroleum pollution in soils. Models based on SVM, PCA, and PLS-DA were applied to discriminate between the spectral contributions of soil samples and those of petroleum and derivatives. In future studies, the predictive models that were developed in the lab can effectively be adapted to detect traces of relevant contaminants in the field. The comparative analyses used are of great importance because they enable the determining of the appropriate statistical tools that should be used to improve the detection of insoluble contaminants in soils and enable better decision making and effectively minimize negative environmental impacts.

The importance of this study is that it can be considered as an alternative method to determine soil contamination due to oil spills quickly. Many of the contamination problems posed by the presence of hydrocarbons in soils are because their detection is neither in situ nor timely. For example, certain soils contain traces of oil even after cleanup and are not monitored because oil concentration is considered low ($\sim$600–20,000 mg·kg$^{-1}$, 0.06–2%) [51,52], rendering them as "decontaminated." In accordance with Dutch legislation (VROM, 1987) [53], there are three limit values of total petroleum hydrocarbons mass per mass of soil: reference (S: 10 mg·kg$^{-1}$; 0.001%), intervention (I: 1000 mg·kg$^{-1}$; 0.1%), and alert (T: 505 mg·kg$^{-1}$; 0.05%). The S value indicates the level at which the soil and groundwater are considered "clean". The I value indicates the level above which it becomes a risk to human health and the environment.

## 2. Materials and Methods

### 2.1. Materials

The insoluble pollutant tested in this study was crude petroleum. The petroleum samples were obtained from a petroleum well and were provided by an oil company in Tame-Arauca (Colombia). Potassium bromide (KBr) was used as a standard reference to identify the oil signals in the presence of particulate materials such as KBr, which does not produce a signal in the MIR region. The KBr was

obtained from Sigma-Aldrich (a subsidiary of Merck KGaA, St. Louis, MO, USA). Sea sand, bentonite, montmorillonite (pillared aluminum clay), and red and brown soils were used as the solid matrices in this study. The sea sand, bentonite, and montmorillonite were purchased from Sigma-Aldrich (St. Louis, MO, USA).

The experiments were conducted in Mayaguez, Puerto Rico, USA. The red and brown soil samples were obtained from 18°9′36″ N, 67°6′40″ W and 18°13′25.7″ N, 67°07′51.2″ W, respectively, in Mayaguez, PR, USA.

## 2.2. Sample preparation

Four dissimilar groups of samples were prepared in the laboratory to mimic soils contaminated by various petroleum concentrations. The first sample group consisted of mixtures of red soil with petroleum concentrations ranging from 0% to 15% *w/w*. The second sample group consisted of mixtures of brown soil with petroleum concentrations ranging from 0% to 10% *w/w*. The third sample group was prepared by mixing sea sand with petroleum concentrations from 0% to 8% *w/w*. The samples in the fourth group consisted of bentonite with 2%, 4%, and 12% w/w petroleum concentrations. Finally, the standard reference samples were prepared using KBr with various petroleum concentrations (e.g., 1%, 2%, 5%, 7%, 10%, and 15% *w/w*). Montmorillonite soil was used as a negative control (0% *w/w* petroleum).

The samples were homogenized by mechanical mixing using a mini vortex mixer for approximately 10 s at 3000 rpm. Next, the mixtures were mechanically mixed again in a shaker mixer. Finally, three different sizes of soil distributions were determined by sieving approximately 50 to 100 g of soil samples with two stacked sieves (USA standard testing sieve, ASTM E-11 specification; ASTM, West Conshohocken, PA, USA) # 20 (hold diameter: 0.850 mm or 0.0331 inches) and # 60 (hold diameter: 0.250 mm or 0.0098 inches). Separate measurements were obtained from soils with diameters of >0.850 mm, 0.850–0.250 mm, and <0.250 mm.

## 2.3. Instrumentation

The QCL spectra were acquired in back reflection using a MIR predispersive spectrometer (LaserScan™, Block Engineering, Marlborough, MA, USA). The spectrometer was equipped with three laser diodes with tuning ranges from 990 to 1111 $cm^{-1}$, 1111 to 1178 $cm^{-1}$, and 1178 to 1600 $cm^{-1}$. The scan time was approximately 0.5 s per diode for a total scan time of 1.5 s. The average power typically varied between 0.5 mW and 10 mW across the 600 $cm^{-1}$ tuning range with 100:1 transverse electromagnetic polarization ($TEM_{oo}$) and a beam divergence of <2.5 mrad on the *x*-axis and < 5 mrad on the *y*-axis. A 3 in. diameter ZnSe lens was used to focus the MIR beam, to collect the reflected light, and to focus the light onto a thermoelectrically cooled mercury-cadmium-telluride (MCT) detector. The wavelength accuracy and precision were nominally 0.5 $cm^{-1}$ and 0.2 $cm^{-1}$, respectively. The spectroscopic system was designed to work best at a target distance of 15 ± 3 cm, with each diode producing an elliptical beam spot of 4 × 2 mm (see Pacheco-Londoño, et al. [54]).

## 2.4. Acquisition of Spectra

The QCL spectrometer was used for reflectance (R) acquisitions across the wavenumber range of 990 to 1600 $cm^{-1}$. The samples were placed in metal holder wells (2.54 cm diameter and 10 mm deep), and the soil surface was flattened to create a smooth surface for accurate measurements. Spectra were acquired at various points on the surface, resulting in 100 reflectance spectra per petroleum concentration in the soil samples. To obtain QCL spectra, an initial background spectrum from a rough gold surface or KBr powder was acquired due to the lack of MIR reflected/backscattered signals from the solid matrices used in this study.

*2.5. Artificial Intelligence and Multivariate Analysis*

MVAs such as PLS-DA and PCA were implemented using the PLS-Toolbox 8.0 (Eigenvector Research, Inc., Wenatchee, WA, USA) for MATLAB ver. 8.6.0.267246 (R2015b; The Math Works Inc. Natick, MA, USA). The results of the PLS-DA modeling were initially evaluated in terms of leave-one-out cross-validation (CV) [55] and external evaluation (Pred). The Pred was accomplished using new samples that were not used in the calibration (Cal) model. Standard normal variate (SNV) was applied as a preprocessing step to all spectral data. The preprocessing allowed the models to provide better results compared to the derivatives and mean centering since SNV eliminates contributions due to variations in the particle size of the different soil types utilized in the study.

The performance of the PLS-DA models was evaluated using the parameters of a confusion matrix, such as sensitivity and specificity of the calibration, validation, and prediction. Sensitivity is the number of samples predicted to be in a class divided by the number of samples in a class, and specificity is the number of samples predicted to not be in a class divided by the actual number of samples, not in a class. The sensitivity and specificity were calculated according to Equations (1) and (2), respectively [50]:

$$\text{Sensitivity} = \text{TP}/(\text{TP} + \text{FN}) \tag{1}$$

$$\text{Specificity} = \text{TN}/(\text{TN} + \text{FP}) \tag{2}$$

where TP, FN, TN, and FP represent the number of true positives, false negatives, true negatives, and false positives. In this work, soils mixed with petroleum and diesel were considered "positive", and soils/sand without added petroleum or diesel were considered "negative." The values for the sensitivity and specificity were calculated for each model, namely, sensitivity (Cal), sensitivity (CV), and sensitivity (Pred), and specificity (Cal), specificity (CV), and specificity (Pred).

SVM was employed using the Logistic Regression CV module to build a classification model using the library of sklearn 3.2 in Python 3. The best hyperparameter was selected by the cross validator, Stratified Kfolds, and was evaluated in terms of CV. The package of libraries in scikit-learn [56] provides a set of open-source software of efficient AI techniques for the Python programming language. They are accessible to non-experts of machine learning and apply to various scientific areas. The external evaluation of the predictions was accomplished using new samples that were not used in the classification model. The criteria that were used to evaluate the performance of the classification model developed with SVM_LoggisticRegressionCV were recall, precision, f1-score, support, and accuracy. In binary classification, recall of the positive class is also known as "sensitivity," and recall of the negative class is "specificity." The precision is also called termed the positive predicted value and is the ratio TP/(TP + FP) where TP is the number of true positives and FP is the number of false positives. The precision (intuitively) is the ability of the classifier of not labeling as positive a sample that is negative. The f1-score is also known as the balanced f-score or the f-measure. The f1-score can be interpreted as a weighted average of the precision and recall, where the f1-score best value is 1 and the worst is 0. The relative contributions of precision and recall to the f1-score are equal. The formula for the f1-score is shown in Equation (3):

$$\text{f1-score} = 2 * (\text{precision} * \text{recall})/(\text{precision} + \text{recall}) \tag{3}$$

The f1-score in the multi-class and multi-label cases is the average of the f1-score for each class with weighting that depends on the average parameter. The supported parameter in the SVM is the number of occurrences of each class in y_true. The accuracy classification score in multi-label classification computes accuracy subset: the set of labels predicted for a sample must exactly match the corresponding set of labels in y_true. The reported averages include the macro average (averaging the unweighted mean for each label) and the weighted average (averaging the support-weighted mean for each label).

Other parameters for the model evaluations were obtained using areas under the receiver operator curve (ROC) plots. These were generated using PLS-DA and SVM to evaluate the "sensor" performance. The ROC plots allow inspection of the fundamental trade-off in the models between TP and FP. This provides much more information than a straightforward accuracy calculation. When comparing two models, the ROC plots clearly show that a curve that is entirely over another represents a model with better results regardless of the threshold used. The area under an ROC plot is equal to the probability that a randomly selected positive case will receive a higher score than a randomly chosen negative case. In other words, it is the probability of sensing the contaminant.

## 3. Results and Discussion

### 3.1. Spectral and PC Analysis

Figure 1 shows representative reflectance spectra of soil matrices used in this study without petroleum added. A KBr/10% *w/w* petroleum standard is also included for reference. All soil spectra, namely, sea sand (sand), bentonite (bent), montmorillonite (mont), and red and brown soils, were acquired without significant difficulty, producing the distinctive signatures in 1000 to 1600 cm$^{-1}$ region. Sea sand samples do not provide any noticeable signal spectral reflectance in 1300 to 1600 cm$^{-1}$ region because the sand samples absorb much of the IR light in this region and a poor reflectance spectrum is produced. Therefore, sand spectra were scaled by a factor of 10 (dotted blue trace) and are included in Figure 1 to provide a profile of sea sand spectral details obtained by QCL from 1300 to 1600 cm$^{-1}$. Nevertheless, sand displays a sharp spectral contrast in the region from 1000–1300 cm$^{-1}$ with an intense signal at 1158 cm$^{-1}$ that is associated with Si-O vibrational stretching mode [57].

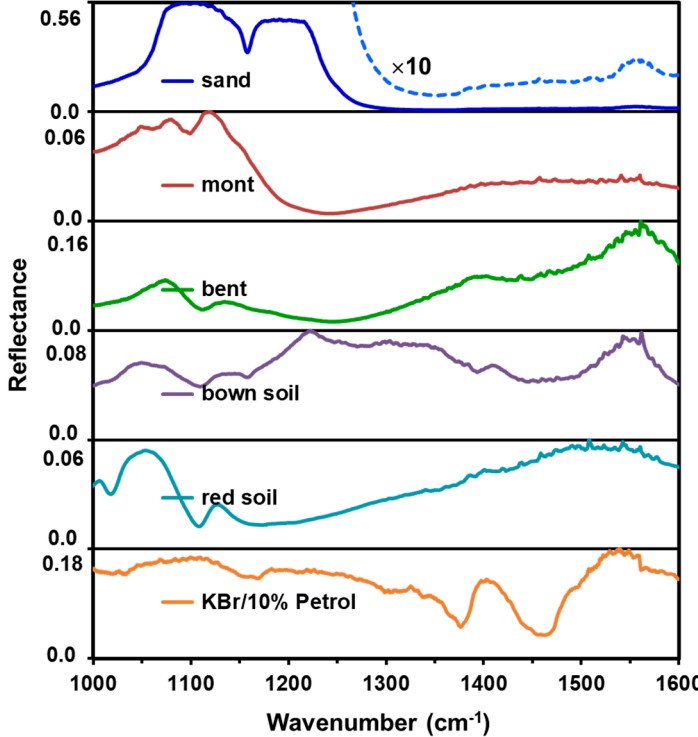

**Figure 1.** Representative reflectance spectra of the various samples without petroleum contaminants: sea sand (sand), montmorillonite (mont), bentonite (bent), brown soil, red soil, and KBr/10%-Petrol reference sample.

Spectral interpretations and band assignments in the fingerprint region (1000 to 1600 cm$^{-1}$) for soil samples are challenging because of the various contributing soil signatures can overlap due to the presence of minerals or to interactions of organic chemicals [58,59]. According to the report by

Janik et al. 1998, most of the peaks below 1300 cm$^{-1}$ can be attributed to clay minerals. In the case of soils with dominant organic matter compositions, strong alkyl peaks (1449 cm$^{-1}$) and strong broad bands due to carboxylic-protein-aromatic species (1750–1449 cm$^{-1}$) can also be present. The standard reference spectrum of 10% petroleum in KBr shown in Figure 1 (solid orange trace) is where the most important petroleum signals are located. The weak signal at 1019 cm$^{-1}$ corresponds to twisted CH$_2$, the weak signal at 1030 cm$^{-1}$ is associated with C–C symmetric stretch. The medium-reflectance signal at 1156 cm$^{-1}$ is attributed to the combination band of CH$_3$ rocking + C–C stretching [60].

For a suitable identification of the signals, it is necessary to apply an ash spectral subtraction to the entire soil spectrum. Performing detection means that the soil to be analyzed does not require any form of physical separation, i.e., the soil remains in its natural way as much as is possible. For this reason, the detection of petroleum traces focused mainly on natural soils, such as red and brown soils, and the addition of synthetic sea sand produced one of the most challenging lands to analyze due to its ability to absorb infrared light. Nevertheless, in an attempt to get closer to in situ detection, this step was omitted.

A PCA model was generated to classify the soil types. The scores plot of principal component 1 (PC 1) vs. PC 2 is shown in Figure 2. PC 1 accounts for 47.27% of the variation and can be used for differentiation between sand and clay. PC 1 also allowed the separation between clayey soil (e.g., mont, bent, and red soil) vs. sandy soil (e.g., brown soil and sand) can also be observed. The signal near 1020 cm$^{-1}$ in red soils was attributed to aluminosilicate lattice vibrations. Besides, the signature near 1100 cm$^{-1}$ noted for mont, bent, and red soils were assigned to the fundamental silicate stretching mode or to (O–Al–O) stretching deformations that produce similar intense reflectance bands. PC 2 can differentiate between the various types of clayish soils and indicates that red soils contain high proportions of bentonite.

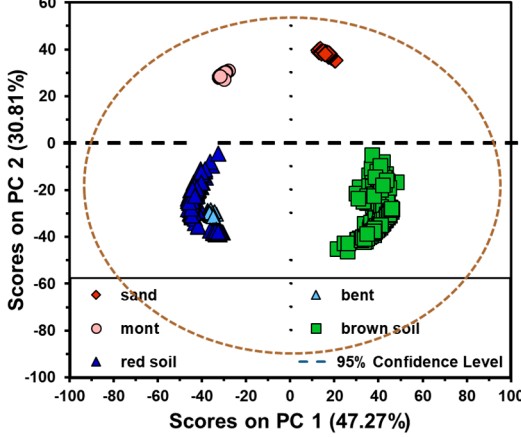

**Figure 2.** Scores plot of PC 2 vs. PC 1 (principal components) for sea sand (sand), montmorillonite (mont), bentonite (bent), brown soil, and red soil. The 95% confidence level is also shown.

Reflectance spectra of samples containing sea sand, red soil, and brown soil mixed with 8% and 10% *w/w* petroleum, are shown in Figure 3. Changes in spectral features for soils and sea sand samples, with and without the pollutant, are visible.

The solid black trace spectra shown in Figure 3a–c represent the reference standard with 10% petroleum in KBr. The spectral signals shown in light blue are from soil samples without petroleum. When these samples are mixed with oil (red solid trace), high intensity bands at 1450 and 1380 cm$^{-1}$ appeared clearly in the spectra (Figure 3b,c). However, as shown in Figure 3a, these signals were feeble (red dotted trace) apparently due to the same reason that mid to high intensity bands are not observed in the spectrum of uncontaminated sea sand. Bands at 1450 and 1380 cm$^{-1}$ are associated with the asymmetric and symmetric deformations of CH$_3$, respectively [61].

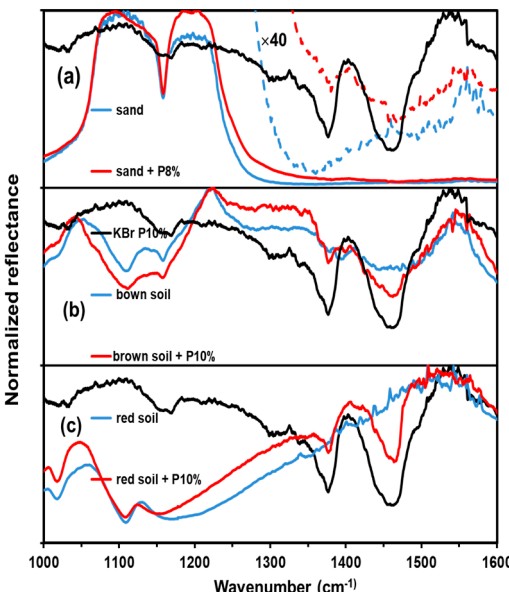

**Figure 3.** Reflectance spectra of soils with and without the pollutant. (**a**) Sea sand + 8% *w/w* petroleum (sand + P8%); (**b**) brown soil + 10% *w/w* petroleum (brown soil + P10%); and (**c**) red soil + 10% *w/w* petroleum (red soil + P10%).

## 3.2. PLS-DA Analysis

The first PLS-DA model was developed using the full spectral region of the excitation of the QCL. Reflectance spectra from 1000 to 1600 cm$^{-1}$ from the various soil types, with and without petroleum, were used, and a PLS-DA model with a relatively poor ability to discriminate sea sand was obtained (results not shown; see Supplementary Material). The ability of this model to detect pollutants in sea sand samples was so poor because of the low intensity of the signals between 1300 and 1600 cm$^{-1}$ due to the low reflectance of the sand samples in this spectral region. Thus, a second PLS-DA model was generated using only the spectral region between 1300 and 1600 cm$^{-1}$ and an improved differentiation was achieved. Figure 4 shows the differentiation between different soil types, with and without petroleum traces, for the improved PLS-DA model. Above the red dotted line (discrimination threshold), the classification of petroleum concentrations ranging from 0.2% to 10% *w/w* in the soil samples was nearly perfect. The PLS-DA model had sensitivity and selectivity values of 0.936 and 0.976 for CV and 0.942 and 0.964 for test validation (TV), respectively. These values are indicative that the model can predict samples correctly in the soils studied. It also indicates that the model is robust, and the technique is robust enough for the detection of petroleum in soil. The classification of some samples was not entirely correct, and there were some failures in discrimination. As shown in Figure 4, most of the discrimination failures were observed in samples of brown soil with low petroleum concentrations (≤0.2%). This was because this type of soil contains high amounts of organic matter. Therefore, brown soils show spectral peaks near 1413 cm$^{-1}$, typical of organic matter carboxylate species that can mislead the discrimination model.

PLS-DA models based on reflectance spectra of the solid matrices and the pollutant concentrations were constructed separately. Models developed based on the solid forms for soils (brown and red soils), and a separate PLS-DA model was developed for sea sand only (models not shown; see Supplementary Material).

A statistical analysis using the parameters of sensitivity, selectivity, and area under the ROC plot was developed to represent the changes in probability for sensing contaminated soils when the petroleum concentrations in soil and sea sand varied, as shown in Figure 5. The sensitivity parameters for Cal., CV., and Pred.; the specificity parameters for Cal., CV., and Pred.; and the area ROC for each PLS-DA model was developed given the petroleum concentrations and types of solid matrices;

these values are listed in Table 1. Additionally, a PLS-DA model using all data was developed for the optimized spectral region between 1300 to 1600 cm$^{-1}$, the area ROC, parameters such as sensitivity and specificity for Cal., CV., and Pred. were also calculated.

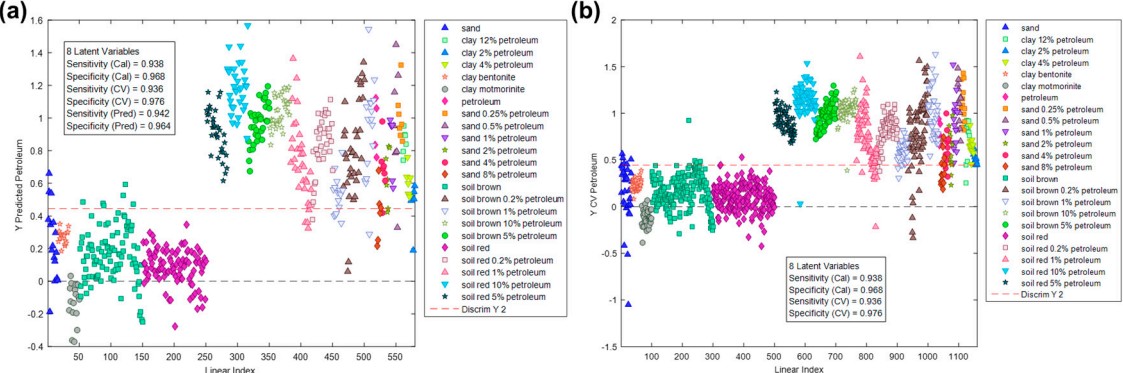

**Figure 4.** Improved partial least square-discriminant analysis (PLS-DA) model for discriminating various soil types using the spectral region from 1300 to 1600 cm$^{-1}$, with and without traces of petroleum. The plot shows y-predicted values by the cross-validation (CV) of samples contaminated with oil vs. the linear index of the samples. (**a**) Test PLS-DA model for the prediction of petroleum; (**b**) CV for the PLS-DA model for the prediction of petroleum.

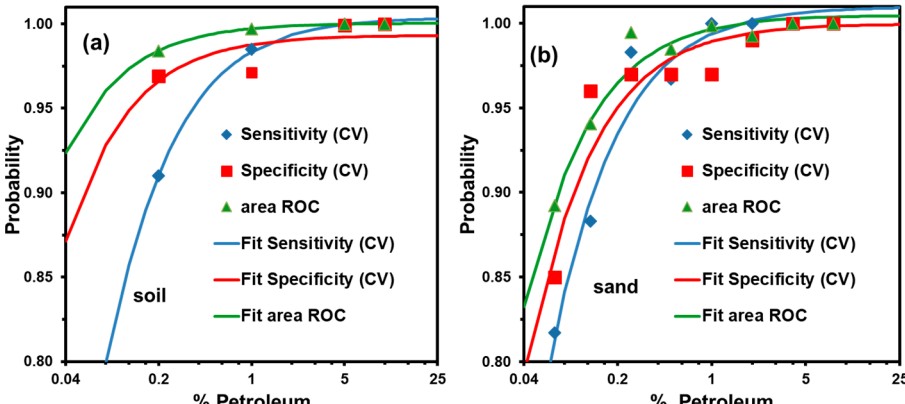

**Figure 5.** The PLS-DA detection probabilities for each concentration in (**a**) sand and (**b**) soil matrices.

The areas under the ROC plots represent the probabilities that a randomly selected positive case will receive a higher score than a randomly selected negative case. To determine the detection and decision limits, fitting was carried out for each probability, and the best fit was obtained with a double reverse fit, according to Equation (4):

$$p = 1/(a + [\, b/\% \text{ of P}])\qquad(4)$$

The petroleum concentrations necessary to accurately measure contamination at probability levels of 50% and 93% for samples were determined from the fit (Equation (4)). These values correspond to the limit of decision (LD) and the limit of detection (LOD), respectively [62]. Table 2 shows the fitting parameters and the prediction % of petroleum for each probability value (50% and 93%). The LD and LOD values for the sand and soil samples were 0.02% and 0.2%, respectively (bold numbers). These were calculated using the fit of the probability of sensitivity (CV) for extrapolation because this is the probability of sensing.

**Table 1.** Sensitivity, specificity, and receiver-operator curves (ROC) area for each PLS-DA model.

| % of P | Sensitivity | | | Specificity | | | ROC Area |
|---|---|---|---|---|---|---|---|
| **Soil** | | | | | | | |
| | Cal | CV | Pred | Cal | CV | Pred | |
| 0.2 | 0.91 | 0.91 | 0.89 | 0.97 | 0.97 | 0.95 | 0.98 |
| 1 | 0.99 | 0.99 | 0.98 | 0.97 | 0.97 | 0.96 | 1.00 |
| 5 | 1.00 | 1.00 | 1.00 | 1.00 | 1.00 | 1.00 | 1.00 |
| 10 | 1.00 | 1.00 | 1.00 | 1.00 | 1.00 | 1.00 | 1.00 |
| **Sand** | | | | | | | |
| 0.068 | 0.85 | 0.82 | 0.77 | 0.87 | 0.85 | 0.80 | 0.89 |
| 0.125 | 0.96 | 0.88 | 0.84 | 0.92 | 0.96 | 0.91 | 0.94 |
| 0.25 | 0.99 | 0.98 | 0.94 | 1.00 | 0.97 | 0.93 | 0.99 |
| 0.5 | 0.99 | 0.97 | 0.94 | 0.97 | 0.97 | 0.94 | 0.98 |
| 1 | 0.99 | 1.00 | 0.98 | 1.00 | 0.97 | 0.95 | 1.00 |
| 2 | 0.99 | 1.00 | 0.99 | 1.00 | 0.99 | 0.98 | 0.99 |
| 4 | 1.00 | 1.00 | 1.00 | 1.00 | 1.00 | 1.00 | 1.00 |
| 8 | 1.00 | 1.00 | 1.00 | 1.00 | 1.00 | 1.00 | 1.00 |
| **All data** | | | | | | | |
| Model-All | 0.93 | 0.92 | 0.91 | 0.93 | 0.92 | 0.88 | 0.94 |
| Model-OPT | 0.96 | 0.96 | 0.94 | 0.96 | 0.96 | 1.00 | 0.99 |

**Table 2.** PLS-DA parameters for the fits for 93% and 50% probabilities for petroleum sensing.

| Sample | Parameter | a | b | R | % for p = 93% | % for p = 50% |
|---|---|---|---|---|---|---|
| Soil | Sensitivity (CV) | 0.996 ± 0.001 | 0.0205 ± 0.0004 | 0.9996 | 0.25 | 0.02 |
| | Specificity (CV) | 1.007 ± 0.009 | 0.006 ± 0.004 | 0.8477 | 0.08 | 0.006 |
| | ROC Area | 0.9996 ± 0.0001 | 0.00332 ± 0.00005 | 0.9998 | 0.04 | 0.003 |
| Sand | Sensitivity (CV) | 0.990 ± 0.008 | 0.016 ± 0.001 | 0.9808 | 0.2 | 0.02 |
| | Specificity (CV) | 1.00 ± 0.01 | 0.010 ± 0.002 | 0.9336 | 0.1 | 0.01 |
| | ROC Area | 0.995 ± 0.005 | 0.0082 ± 0.0008 | 0.9576 | 0.1 | 0.01 |

## 3.3. SVM Analysis

Unlike the PLS-DA models generated, the SVM model was developed using the full reflectance spectral region (e.g., from 1000 to 1600 cm$^{-1}$) with all soil types, including sea sand with and without petroleum. The classification model used petroleum concentrations ranging from 0% to 12% *w/w* in all soil samples. Additionally, a second model combining PCA and SVM was introduced. The PCA-SVM model was carried out to establish an SVM model with reduced data dimensions, and six PCs were used. An external evaluation of the classifications for the PCA-SVM and SVM models was almost perfect and is shown in Table 3. The PCA-SVM model has a recall value of samples with petroleum (sensitivity) and recalls value for samples without oil (specificity) of 0.93 for both cases. However, the SVM model generated using all data provided improved results with recall values (sensitivity and specificity) of 1.00 and 0.99, respectively. Table 3 shows the accuracies of the evaluation methods performed on the test sets, and the table data show that the SVM model achieves better accuracy rates when compared with the PCA-SVM model.

**Table 3.** Parameters for the external evaluation for each support vector machine (SVM) model.

| | Test PCA-SVM | | | |
|---|---|---|---|---|
| | **Precision** | **Recall** | **f1-score** | **Support** |
| Without P | 0.89 | 0.93 | 0.91 | 127 |
| With P | 0.95 | 0.93 | 0.94 | 191 |
| Accuracy | | | 0.93 | 318 |
| Macro avg. | 0.92 | 0.93 | 0.93 | 318 |
| Weighted avg. | 0.93 | 0.93 | 0.93 | 318 |
| | **Test SVM** | | | |
| | **Precision** | **Recall** | **f1-score** | **Support** |
| Without P | 0.99 | 1.00 | 0.99 | 127 |
| With P | 1.00 | 0.99 | 1.00 | 191 |
| Accuracy | | | 1.00 | 318 |
| Macro avg. | 0.99 | 1.00 | 1.00 | 318 |
| Weighted avg. | 1.00 | 1.00 | 1.00 | 318 |

The LD and LOD were calculated from the ROC chart (see Figure 6a) using the best model developed by the SVM, and the results are summarized in Table 4. The behavior of the probability of sensing the contaminant using SVM was evaluated by using the equation 4 fitting values of the LOD and LD, which were 0.2% and 0.01%, respectively. These values are below those found in restored soils (2%).

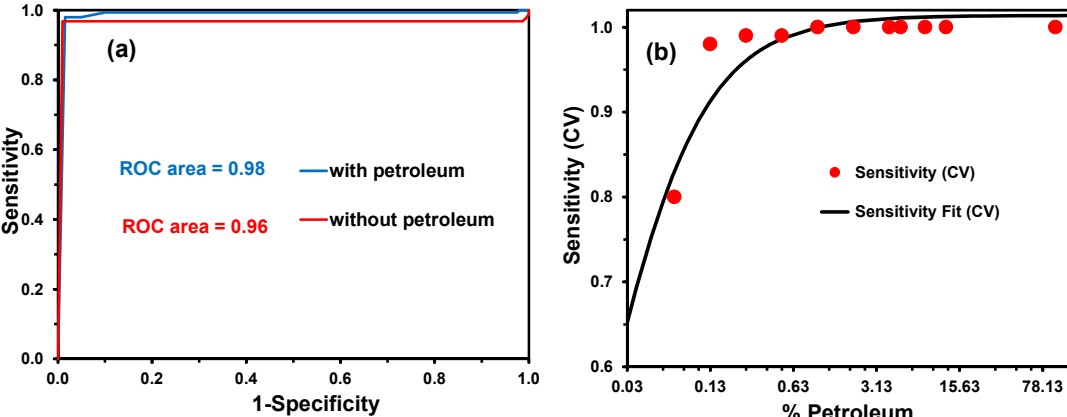

**Figure 6.** (**a**) ROC curve from the quantum cascade laser/ support vector machine (QCL/SVM) model based on a sigmoid kernel function and (**b**) SVM detection probabilities for each concentration in all soil matrices using the SVM model.

**Table 4.** SVM parameters for the fits and the 93% and 50% probabilities for petroleum sensing.

| Parameter | a | b | R | % for p = 93% | % for p = 50% |
|---|---|---|---|---|---|
| Sensitivity (CV) | 0.99 ± 0.01 | 0.014 ± 0.002 | 0.923 | 0.2 | 0.01 |

To analyze the trade-off between accurate detections and false alarms in the QCL/SVM model, a ROC plot was generated (Figure 6a). As can be observed, the ROC plot that produces the best possible prediction yields a point in the upper left corner, i.e., coordinate (0,1) in the ROC space, representing 100% sensitivity (no false negatives) and 100% specificity (no false positives).

The area under a ROC plot represents the probability that a randomly selected positive case will receive a higher score than a randomly chosen negative case; in other words, it is the probability of sensing. From the fit using Equation (4), the LD and LOD were calculated (see Figure 6b).

## 4. Conclusions

This study demonstrated that new spectroscopic techniques, such as MIR laser spectroscopy, combined with PLS-DA and SVM, can be used for the development of practical methods for in situ detection in challenging matrices, such as sensing petroleum traces in soils. Powerful multivariate routines, such as PCA, were applied to reduce the number of primary variables and to separate the spectral signals from different solid matrices (e.g., clay soils and sandy soils). The reflectance signals of petroleum at 1450 and 1380 cm$^{-1}$ associated with the asymmetric and symmetric deformation of CH$_3$ were easily identified in the spectra from soil mixtures. However, the identification of petroleum signals in sea sand samples required a spectral zoom of 40 times. The detection of petroleum in sea sand was much more complicated due to the sea sand samples which absorb MIR radiation profusely in the region between 1300 to 1600 cm$^{-1}$. This was solved by generating specific discriminant models for each soil type. The proposed QCL/PLS-DA methodology was successfully tested and used on laboratory samples and provides a quick and simple way to analyze soils contaminated with petroleum. The methodology can be adapted for field applications.

Although the experiments were limited to only two types of particulate materials (whole soils and sand), the technique is promising for detecting petroleum contamination based on the results obtained. Future work can involve expanding the models to other soil types and possibly to use other robust pattern recognition techniques.

The QCL/SVM methodology demonstrated superior performance because a nonlinear classification was implemented. For low petroleum concentrations, the behavior was not linear, and for this reason, this model included these points. The SVM model improved the detection and decision limits and yielded extrapolated values of 0.02% and 0.01%, respectively. Even though an extrapolation procedure calculated these values, their certainty probability is lower than for QCL/PLS-DA. However, the QCL/SVM methodology had smaller values of LOD (0.25%) and LD (0.02%).

**Supplementary Materials:** The following is available online at http://www.mdpi.com/2076-3417/10/4/1319/s1. Figure S1. PLS-DA model for calibrating the discrimination of several soil types with and without petroleum traces, using the spectral region from 1000 to 1600 cm$^{-1}$. Figure S2: Improved PLS-DA model for calibrating the discrimination of various types of soils, with and without petroleum traces, using the spectral region from 1300 to 1600 cm$^{-1}$. Figure S3: PLS-DA model calibrating the discrimination of red and brown soils containing 0.2% petroleum and without petroleum traces in the spectral region of 1000 to 1600 cm$^{-1}$. Figure S4: PLS-DA model for calibrating the discrimination of red and brown soils containing 1% petroleum and without petroleum traces in the region of 1000 to 1600 cm$^{-1}$. Figure S5: PLS-DA model for calibrating the discrimination of red and brown soils containing 5% petroleum and without petroleum traces in the region of 1000 to 1600 cm$^{-1}$. Figure S6: PLS-DA model for calibrating the discrimination of red and brown soils with 10% petroleum and without petroleum traces in the region of 1000 to 1600 cm$^{-1}$. Figure S7: PLS-DA model for calibrating the discrimination of sands with 0.0625% petroleum and without petroleum traces in the region from 1000 to 1600 cm$^{-1}$; Figure S8: PLS-DA model for calibrating the discrimination of sands with 0.125% petroleum and without petroleum traces in the region of 1000 to 1600 cm$^{-1}$. Figure S9: PLS-DA model for calibrating the discrimination of sands with 0.25% petroleum and without petroleum traces in the spectral region from 1000 to 1600 cm$^{-1}$. Figure S10: PLS-DA model for calibrating the discrimination of sands with 0.5% petroleum and without petroleum traces in the spectral region from 1,000 to 1,600 cm$^{-1}$. Figure S11: PLS-DA model for calibrating the discrimination of sands with 1% oil and without petroleum traces in the spectral region from 1000 to 1600 cm$^{-1}$. Figure S12: PLS-DA for calibrating the discrimination of sands with 2% petroleum and without petroleum traces in the spectral region from 1000 to 1600 cm$^{-1}$. Figure S13: PLS-DA model for calibrating the discrimination of sands with 4% petroleum and without petroleum traces in the spectral region from 1000 to 1600 cm$^{-1}$. Figure S14: PLS-DA model for calibrating the perception of sands with 8% oil and without petroleum traces in the spectral region from 1,000 to 1,600 cm$^{-1}$.

**Author Contributions:** Conceptualization, N.J.G.-F.; Data curation, M.L.O.-C. and A.R.M.-G.; Formal analysis, N.J.G.-F. and M.L.O.-C.; Funding acquisition, S.P.H.-R.; Investigation, L.C.P.-L.; Methodology, N.J.G.-F., M.L.O.-C. and A.R.M.-G.; Project administration, S.P.H.-R.; Resources, S.P.H.-R.; Software, R.V.-G.; Supervision, L.C.P.-L.; Validation, R.V.-G.; Visualization, R.V.-G. and L.C.P.-L.; Writing—original draft, N.J.G.-F. and L.C.P.-L.; Writing—review & editing, S.P.H.-R. and L.C.P.-L. All authors have read and agreed to the published version of the manuscript.

**Funding:** This research received no external funding.

**Acknowledgments:** The authors thank the Center for Chemical Sensors (CCD), and the Chemical Imaging and Surface Analysis Center (CISAC) for the QCL instrumentation and MacondoLab business growth center for artificial intelligence detail. The authors thank the members of Macondo Lab Edgar Medina-Ahumada by the help in the Graphical Abstract design and Juan Pestana-Nobles by the assistance in the implementation of SVM in Python.

**Conflicts of Interest:** The authors declare no conflict of interest.

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
