# Peer review of "Artificial Intelligence Assisted Mid-Infrared Laser Spectroscopy In Situ Detection of Petroleum in Soils"

_applsci, doi:10.3390/app10041319_

Round 1

Reviewer 1 Report

The authors of "Quantum Cascade Laser Assisted by Artificial Intelligence for Standoff Detection of Petroleum Traces in Soils" describe the implementation of a reflectance spectrometer with a quantum cascade laser for the use of in-situ detection of petroleum in several soil types.  Additionally, the authors leverage machine learning algorithms for the determination of petroleum content in soil samples.

Overall, the design of the experiment is good, and the results are of reasonable quality.  There are several points that the authors should address before publication in Applied Sciences.

1) Calling this detection technique "standoff detection" may not be appropriate. When acquiring spectra, the authors noted that the soil samples needed to be flattened and that the laser source was placed roughly 15 cm from the sample.  While this is an in-situ measurement, it does not seem that this is a "standoff" measurement as the measurement takes place very close to the sample, and there is a minor sample preparation needed. 

2) The "fit" shown in figure 6(b) is extrapolated to get a sensitivity of 0.04% of petroleum.  While the fit generally follows the data (on a logarithmic scale), extrapolation to lower % petroleum seems dubious.

3) Related to figure 6(b), there is very little separation in the sensitivity of the % petroleum shown (0.98 to 1.00).  Therefore, it would be advantageous to have additional samples with lower % petroleum to demonstrate a larger deviation in CV.

4) The authors note in the introduction that there currently is a problem in the remediation of oil sites because oil contamination is not evident at first sight, and/or that certain soils are not monitored because the oil concentration is low, and the soil is considered decontaminated.  What level (%) of petroleum/oil is this a problem for?  How does it compare with the detection limit discussed in the manuscript?

There are several typesetting and grammatical errors throughout the paper as well.  It is advised that the manuscript should be thoroughly proof-read.

A few examples:

Formatting of cm^{-1}.  the - sign and 1 do not align as subscripts throughout the manuscript.

line 203: A sensitivity and specificity were calculated for each model (as opposed to three sensitivities and specificities calculated for each model)

A typographical error in the x-axis label of figure 6(b).

Reviewer 2 Report

Your paper  deserves publication after several improvements as outlined to be made still.

The Quantum Cascade laser needs to be described more carefully, its performance defined and the relevance for Petroteum traces more carefully defined. Where can one purchase this facility?

The results obtained should be applicable to users. Therefore your Figure 4 needs to be more carfully extended for examples of applications for your experimental approach.

Can you provide more details of the Petroleum employed such as electrical conductivity and spectral data as obtained possibly by THz spectroscopy?
